# A Simple and Effective Method for Measuring the Density of Non-Newtonian Thickened Tailings Slurry during Hydraulic Transport

**DOI:** 10.3390/s22207857

**Published:** 2022-10-16

**Authors:** Maciej Filip Gruszczyński, Stanisław Kostecki, Szymon Zieliński, Zbigniew Skrzypczak, Paweł Stefanek, Stanisław Czaban, Marcin Popczyk

**Affiliations:** 1Institute of Environmental Engineering, Wrocław University of Environmental and Life Sciences, Grunwaldzki Sq. 24, 50-363 Wrocław, Poland; 2Faculty of Civil Engineering, Wrocław University of Science and Technology, Wybrzeże Wyspiańskiego 27, 50-370 Wrocław, Poland; 3KGHM Polska Miedź S.A. Oddział Zakład Hydrotechniczny, Polkowicka 52, 59-305 Rudna, Poland; 4Faculty of Mining, Safety Engineering and Industrial Automation, Silesian University of Technology, St. Akademicka 2, 44-100 Gliwice, Poland

**Keywords:** non-Newtonian fluids, density measurement, dredging measurements, densitometers, U-tube densitometer

## Abstract

The extension of the Żelazny Most tailings store facility (TSF), which is the largest in Europe, requires the transport of large amounts of tailings, e.g., from the central area of the TSF reservoir to the new southern extension (SE). In order to use the mature fine tailings deposits located under the clarified water in the TSF reservoir, which were thickened in the process of natural sedimentation, it was decided to choose suction dredgers that collect tailings a few meters from under the water surface. The dredgers, which are most commonly used for the extraction of sand or gravel, showed the ability to pump fine flotation tailings slurry in the conducted tests. However, in order to reduce the costs of the entire operation, it was necessary to control the density of the slurry. The article presents a prototype installation for measuring the efficiency of the solid phase of the “in situ” dredging process in real time. The installation was designed for the needs of dredging a deposit of tailings that were thickened in the natural sedimentation process, and which had a particle size of below 63 μm. The installation consists of a flow meter, a densimeter, and a section for measuring the head loss of the flow of the slurry. The applied methodology allows for the current assessment of the dredger’s operating parameters, which translates into a more effective–in terms of process efficiency–management of the dredger’s work.

## 1. Introduction

One of the most important aspects of sustainable development strategy in deep mining is the management of tailings. The expansion of landfills and the increased risk of their negative impact on the environment make it necessary to search for new safe storage technologies. Mine tailings are a by-product of the technology of obtaining valuable elements from geological, usually rocky, deposits. Rock grinding, chemical processes, and water rinsing result in sediment that consists of a mixture of water and solid particles with a diameter of 2–100 μm. The tailings are then transported using pipelines to a storage site, which is usually a large-capacity tailings storage facility (TSF). Such a facility must contain over 80% of the produced tailings from many years of operation of mining plants, as well as significant amounts of the water that is used for hydrotransport. Over time, as the reservoir becomes full and the mine’s lifetime extends, it is possible to increase the height of its dams. The storage of tailings, related to the continuous expansion of the reservoir, is a significant cost for mining plants, and therefore the dams are usually elevated with the use of tailings produced in the enrichment process [1]. This is the cheapest way of managing tailings. Sediments that are deposited on the internal slopes of the reservoir flow downwards and create slightly sloping beaches made of thicker fractions of tailings. In the central part of the reservoir, the sediments are subjected to a consolidation process, with the recovered water being reused in the flotation process, or disposed of to the natural environment. In order to reduce the costs and the risk of a catastrophe of the constantly built-up dams of the reservoir, technologies for the compaction of tailings before their landfilling are constantly being sought [2]. Tailings can be thickened with the use of hydrocyclones or gravity thickeners, but this is a technology that requires pumps with high performance (head and flow rate), which in turn is expensive due to the significant demand for energy and chemical agents that support the process (flocculants) [3,4,5,6]. The advantage of depositing highly concentrated sediments is that there is no need to build high dams around the landfill, and also that there is a possibility of its quick recultivation.

An alternative to mechanical thickening in hydrocyclones is the technology of “relocating” the consolidated tailings in the main reservoir to a prepared area where the sediments are further consolidated. It is assumed that their spreading into the landfill will be limited, and therefore the landfilling will have lower energy costs, lower water consumption, greater landfill safety, and a lower environmental impact. The “relocation” can be done using mechanical or hydraulic transport. The latter can bring several advantages, because when compared to mechanical transport, it is dust-free, takes up much less space, is more mobile, enables full automation, and requires a minimum number of operating personnel. However, it also requires the use of high-quality pumping equipment and a control system [7]. Such a mobile method of transporting consolidated tailings is the dredging process, which consists of taking sediment from the bottom of the reservoir with the use of a pump, and then pumping it to a storage site. One of the key elements of this process is the controlled measurement of the excavated deposit’s density and capacity. The measurement of the density and the flow rate allows for the proper control of the dredger’s operation, and prevents the collection of over-hydrated sediments (which are not worth transporting), as well as over-consolidated sediments that can block the pump or discharge line. In addition, the measuring device in the set with flow meter can be used to signal the system’s failure.

The aim of this work is to present a simple technique for measuring the density of sediment transported under pressure through a pipeline. The technique was tested on the Żelazny Most reservoir in southwest Poland, which is a landfill for post-flotation tailings from a copper mine. The article presents a prototype installation for measuring the efficiency of the solid phase of the “in situ” dredging process in real time. The installation was designed for the needs of mining a deposit of mining tailings that were thickened in the process of natural sedimentation, and which have a particle size of below 63 μm.

## 2. Study Site

In southwestern Poland, there are significant deposits of copper ore that have been exploited for many years. In the enrichment process, ground ore is deprived of desired raw materials, while the remaining waste rock, in the form of a tailings slurry, is deposited into the Żelazny Most tailings store facility. Twenty-nine million tons of flotation tailings, with the addition of significant amounts of water, are transported annually by pipelines to the “Żelazny Most” TSF, the active area of which is 1400 ha. Its total capacity is approx. 800 million m^3^. Only a small part of it, about 8 million m^3^, is water that has accumulated in the central part of the reservoir, which, after clarification, is then reused in the copper production process. The remaining volume is filled with sediments. The earth dams around the landfill are 40 to 70 m high, depending on the topography, and their height is constantly increasing due to the continuous depositing of transported sediments. Three quarters of the sediments are deposited onto beaches, where they run off towards the center of the reservoir and segregate [8]. The beach slopes towards the water, which is due to the characteristics of the tailings and the adopted methodology of the depositing process [9]. In the beach zone, in a strip of about 200 m from the crest, the thickest grained material (with the parameter d50 = 0.1 mm on average) is deposited, and further away the value of d50 drops to d50 = 0.01 mm under the water. The tailings that are collected closest to the dams are used to form successive levels of the embankment (each 2.5 m high). About 25% of the finer grained tailings from one of the ore enrichment plants are dumped directly into the TSF reservoir (bypassing the beaches) in order to seal its bottom. As the height of the dams increases, the risk of TSF failure also increases, and therefore in order to keep the risk as low as possible, it was decided to allocate an additional area on the southern side of the reservoir for an extra sediment storage site (Figure 1). In the southern extension (SE), tailings with a higher density than in the TSF will be stored in order to efficiently use its lower capacity.

In the course of many years of analyzes and tests, a decision was made to artificially divide the stream of tailings into coarse-grained tailings, which are intended for the superstructure of the TSF dams, and fine tailings, which are meant to be stored in the central part of the TSF. The division of particles with regards to their size takes place in hydrocyclones. Fine particles, together with the majority of water from the overflow, are characterized by a low concentration and require further compaction, while the underflow is already characterized by a significant density of about 1.5 kg m^−3^ and is pumped to the SE to be deposited in its outer ring (Figure 2).

Fine tailings require the use of expensive and large (with a diameter of several dozen meters) gravity high-rate thickeners that are supported by a flocculant, which are meant to combine fine particles into groups. The sediment particles that are in groups fall to the bottom of the thickeners, and water is then poured through the thickener’s crest. The bottom outlet of the thickener consists of fine condensed tailings that must be pumped (often as a non-Newtonian slurry) to the final deposition site [10].

An alternative solution to sorting sediments in waste thickeners (for the purpose of their storage in the SE) is the transport of tailings deposited in the TSF reservoir. It was assumed that with the passage of time, fine-fraction sediments that are accumulated in the central part of the basin, which can be transported to the SE, thicken by themselves. Therefore, an attempt was made to use a suction dredge, which is normally used to extract sand from under water, and pump the tailings to a storage site. The first tests, starting in 2010 and continuing in the following years, did not confirm concerns about difficulties during pumping, including the possibility of deposits becoming blocked in the dredge pipe. It was found that the sediment consisted of fine particles does not segregate. On the contrary, its physical parameters are highly homogeneous. However, the key to the success of the transport process is the correct, uninterrupted measurement of the density of the pumped medium, which should contain as little water as possible for the purpose of depositing it in the SE. It was decided that sediments with the smallest particle diameter, which lay under the water layer in the deepest parts of the reservoir, would be dredged. However, their mechanical compaction is associated with the highest costs.

The mineralogical characteristics of the post-flotation tailings are as follows: the average specific grain density of the silty and clay fractions is 2.7 g/cm^3^. The chemical composition of the tailings from the Polish copper ore depends on the place of their sedimentation. In the upper part of the beaches, on which the tailings are poured, coarse-grained fractions that are used to erect the body of the dams are deposited. They contain about 70% silica, 8% calcium oxide, 4% alumina, 3% magnesium oxide, 1.5% potassium oxide, and other substances, whereas the loss on ignition amounts to 11%. In the case of the streams of the tailings that are poured directly to the center of the landfill, the silica content is about 30%, while the calcium oxide content increases to 22%, the magnesium oxide content increases to 9%, and the loss on ignition increases to 27%. The mineralogical composition of the tailings with a predominance of sandstone is about 60% quartz, 20% dolomite, 10% muscovite, 5% calcite, 4% gypsum, and about 1% chlorite. In the carbonate ore, the quartz content drops to 50% in favor of calcite and gypsum [11].

## 3. Density Measurements of the Thickened Tailings Slurry

### 3.1. Methods of Measuring Density

The selection of the technique for measuring the density of the medium depends on technical and economic parameters. A typical dilemma is the selection of a method that is reliable and precise, and which has an acceptable investment outlay. Depending on the purpose of the measurement, methods characterized by high measurement reliability over time, high measurement accuracy, or low cost can be considered. Very often, one of the selection criteria is the use of non-invasive methods (e.g., abrasive, harmful media) or methods that do not adversely affect the sediments, and thus the environment (methods with a radioactive source). In the case of large industrial investments, measurement reliability is the leading factor, while in the case of low-cost projects, measurement with the use of expensive equipment, which additionally requires specialized maintenance, is unacceptable. Table 1 lists some of the most popular measurement methods that are used in the mining industry.

The accuracy of the presented methods mainly depends on the place of construction, the parameters of the medium, and the size of network elements: pipe diameters, wall thickness, material, linings, etc. In the case of appropriate working conditions, these methods give errors of up to a few percent, or much below a percent, e.g., the Coriolis method. However, these values must be individually analyzed each time, because the structure of the network and the tested medium have a significant influence on the accuracy of the measurement. As a supplement to the main measuring equipment, some authors propose the use of other parameters of the dredging process as the basis for density determination [33].

Due to the adoption of the method of dredging the sediments from the TSF to the SE in the discussed test studies, a decision was made to use the pressure method of density measurement due to its simplicity, reliability, the possibility of installing a measuring device on a ship’s deck, and its resistance to unfavorable working conditions (including a significant salinity of water in the TSF and variable weather conditions). The lack of negative impacts on the environment and the need to reduce costs were also of great importance when selecting the method. For this reason, the more popular measurement methods, which are based on a radioactive source of radiation, were abandoned.

### 3.2. Measuring Installation and Procedure

A hydraulic densimeter was used to measure the density of the tailings slurry flowing in the pressure pipeline of the dredger. The principle of operation of the device is based on the measurement of the pressure drop on two vertical sections of the pipeline of the same length and diameter [34]. The pressure drop in the measuring section can be described by the principle of conservation of energy:(1)$$-\frac{dP}{dx}=\rho_{m}g\frac{dh}{dx}+\frac{2fV^{2}\rho_{m}}{D}$$
where: *ρ_m_*—slurry density (kg/m^3^); *f*—friction factor estimated from the viscosity and density of the fluid, *D*—inside dimeter of the measuring section (m); *dP*—pressure drop (Pa); *g*—acceleration due to gravity (m/s^2^); *V*—mean velocity of pipe flow (m/s); *h*—elevation above a datum (m).

The device requires the following conditions: (i) the lengths of both measuring sections must be the same and the sections must be arranged vertically, (ii) the minimum distances from the bends of the installation to the beginning of the measuring section must be kept, with the minimum distance depending on the turbulence of the flow, (iii) the phase slip between the solids and the flowing liquid cannot be too large (this applies to a hydromixture with a low density), (iv) the flowing medium should have the same density in both measuring sections.

In the research, it was decided to use a hydraulic densimeter installed on board a vessel (Figure 3). The densimeter was supplied with the tested medium under pressure in the discharge line of the dredger. The flow through the installation was due to the pressure difference between the beginning of the discharge line and the free outflow from the densimeter onto the deck of the vessel. The pressure difference between the inlet and outlet to/from the density measuring installation ranged from 0.2 to 0.35 MPa. The density meter was constructed of translucent PVC pipes with DN 25. The total length of the installation was about 10 m, while the length of the vertical measuring section was 2 × 1.83 m. Two APR-200 HLW type C differential pressure transducers (with a measuring range of 0–250 kPa), equipped with a display, were installed. The impulse tubes supplying pressure to the differential pressure transducer are made of a flexible transparent pipe, and it is therefore possible to control the patency of the tube and to observe any possible air entrainment. The results of the measurements from the pressure transducers were recorded in electronic form with a frequency of 1/3 Hz by the data logger. The conduit was connected by shielded cables with transducers in order to avoid disturbances during the data transmission from the measuring instruments. The installation was also equipped with a positive-displacement pump for rinsing the impulse tubes and venting the measuring system. The impulse tubes, which were filled with water, were attached at points 1–4. The pressure sampling points were made of triple connectors installed at the beginning and end of the measuring section. The measured pressure was transferred to the diaphragm of the differential pressure transducers, which were installed on each of the densimeter arms.

By transforming (1), the operating principle of the applied densimeter can be described by Equation (2):(2)$$\rho_{m}=\frac{\left(P_{1}-P_{2}\right)+\left(P_{4}-P_{3}\right)}{2gL}$$
where: *P_i_*—pressure at point *i* (Pa); *L*—length of measuring section (m).

Due to the applied differential pressure transducers, instead of measuring the pressure at points 1–4, Equation (2) may take the form written as for the pressure difference (3):(3)$$\rho_{m}=\frac{\Delta P_{1-2}+\Delta P_{4-3}}{2gL}$$
where: *∆P_i-j_*—hydrostatic pressure difference in the vertical arm of the densimeter between points *i* and *j* (Pa).

The use of differential pressure transducers instead of pressure transducers allowed the accuracy of the measurements to be increased. This is because the measurement error is committed twice during one measurement instead of four times. The pressure difference between points 1 and 2 and points 4 and 3 is influenced by two components—the pressure difference resulting from the hydrostatic distribution in the medium flowing through the measuring sections, and the pressure difference resulting from the flow’s resistance. The components of the pressure difference can be written in the form of Equation (4). The component of the flow’s head loss of the pressure difference between points 1 and 2 has a positive sign, while between points 4 and 3 it is negative.
(4)$$\Delta P_{1-2}=\Delta P_{1-2}^{hp}+\Delta P_{1-2}^{hl}\;\;\;\;\Delta P_{4-3}=\Delta P_{4-3}^{hp}-\Delta P_{4-3}^{hl}$$

By substituting the components of the individual differences from Equation (4) to Equation (3), and by simplifying the component of the flow’s resistance, Equation (5) describing the operation of the device is obtained:(5)$$\rho_{m}=\frac{\Delta P_{1-2}^{hp}+\Delta P_{4-3}^{hp}}{2gL}$$
where index *hp* is the hydrostatic pressure difference in the vertical arm of the densimeter, and index *hl* is head losses.

The test installation of the hydrotransport of the slurry included a dredger for the mining of tailings collected under the surface of the sludge water in the TSF, and a section of the discharge pipe, which was used to determine the amount of hydraulic losses during the flow. The dredge spoil was deposited again to the reservoir or pumped through a steel pipeline to the beach of the facility, from where it flowed down gravitationally towards the reservoir. The scheme of the system, which shows the depositing of the dredge spoil on the beach, is shown in Figure 4. It was decided that the sediments with the smallest particle diameter, which lay under the water layer in the deepest parts of the reservoir, would be dredged, as their mechanical compaction is associated with the highest costs.

## 4. Results and Discussion

The measurement data were used to test the effectiveness and accuracy of the pressure-based measuring device by comparing it with the results of manual measurements made using the volumetric method. The measurement results were recorded in the data logger and included the date and time of the measurements (with an accuracy of 0.001 s), the pressure values on both transducers, and the value of a control flow measurement (obtained with the use of an electromagnetic flow meter). All the data were recorded simultaneously with a frequency of 1/3 Hz, and therefore a record of 6134 samples was made. The density (determined using Formula (5)) of the sediments transported within the 1st measurement campaign is shown in Figure 5.

The density of the transported sediment was approximately 1.3 × 10^3^ kg/m^−3^ (solid black line). However, periods of significant disturbance of the flow, or its readings, were noticeable. Longer periods of disturbance, during which the density values were equal to 1.0 × 10^3^ kg/m^−3^ (dark-gray dashed line), are associated with the stopping and shifting of the suction head (which caught the water above the sediments), and the restarting of the dredger. Minor temporary disturbances in density are due to interruptions in the measurements, which result from the clogging of the measuring pipelines, the necessity to rinse them, and the instability of the flow (temporary frictional resistance) in the discharge pipeline. The analysis of individual types of disturbances was performed by comparing the readings of the hydraulic densimeter with both the results of measuring the flow of the dredge spoil and the notes made by the authors of the study during the measurements (presented in Table 2).

In order to correctly interpret the indications of the densimeter, the results were subjected to statistical processing. In the first step, the measurement values that were diagnosed as outliers were removed (in accordance with Table 2), which resulted in the obtaining of the correct series of density values. The measured values are characterized by high-frequency pulsations and slight changes over longer periods of time (Figure 6). The latter are due to a different density value of the dredge spoil, and their value was determined by smoothing the plot using a wavelet filter and the Block James–Stein estimator [35]. As shown in Figure 6, the filtered density plot of the sediments is mostly smooth and in line with the values that were measured by the manual method. The Pearson correlation coefficient *r* for the original signal *ρ* and the filtered $\hat{\rho }$ was determined from the following formula:(6)$$r=\frac{cov\left(\rho ,\hat{\rho }\right)}{Sd_{\rho }\cdot Sd_{\hat{\rho }}}\;,\;cov\left(\rho ,\hat{\rho }\right)=\frac{\sum_{i=1}^{n}\left(\rho_{i}-\langle \rho \rangle \right)\left({\hat{\rho }}_{i}-\langle \hat{\rho }\rangle \right)}{n-1}$$
where $\rho ,\;\hat{\rho }$ denotes the original and filtered density signal over time, respectively; $\langle \rho \rangle ,\langle \hat{\rho }\rangle$ are the average values; $Sd_{\rho },Sd_{\hat{\rho }}$ are the values of the standard deviation, and *n* is the number of density measurements. The correlation coefficient reached the value of 0.97, which indicates that the filtered signal accurately represents the real measured signal [36].

The sediment density $\hat{\rho }\left(t\right)$ after smoothing using a wavelet filter changes within the range of 1.186–1.389; the average value of the original and filtered signals is the same and amounts to $\langle \rho \rangle =\langle \hat{\rho }\rangle =1.265$ kg/m^−3^; the standard deviation of the original signal is equal to $S_{\rho }=0.021$; and the smoothed signal is equal to $S_{\hat{\rho }}=0.020$. In order to investigate the nature and possible causes of the pulsations, their values were isolated by subtracting the smoothed value $\hat{\rho }\left(t\right)$ from the instantaneous values of the density ρ(t). The obtained pulsation diagram is shown in Figure 7.

The value of the density fluctuation varies within ±0.02 × 10^3^ kg/m^−3^. In order to check whether the fluctuations do not come from the vibrations of the measuring system or the transmission pipeline, an analysis of their frequency using the FFT method was performed. The frequency analysis graph is shown in Figure 8.

The signal in Figure 8 shows the lack of a dominant frequency, and it can therefore be concluded that the density pulsations are not the result of vibrations of the measuring system, with their origin instead resulting from the variability of the instantaneous shear resistance of the sediments in the measuring pipeline.

The verification of the measurements with the use of the hydraulic densimeter was carried out on the basis of the measurements made using the manual volumetric method, which involves the taking of the sediment sample directly from the pipeline of the measuring apparatus. Manual density tests were carried out eight times during the entire dredging process and compared with the results of density measurements made using pressure apparatus. The comparison is shown in Table 3 and Figure 6 (blue dots).

The mean absolute error values do not exceed 2.5% (MAE = 0.018 Mg/m^−3^), and therefore it can be concluded that the presented apparatus for measuring density using the pressure method is sufficiently accurate. The reasons for the discrepancy between the two measurements may be due to the time difference resulting from the duration of the period of performing both types of measurements. Measurements with the use of the apparatus are instant, and when using the manual method, they lasted about 15 s. During this time the manometer readings were withheld, in order not to take into account disturbances caused by collecting the sediment from the measuring system (see Figure 3).

## 5. Conclusions

This paper presents a simple and effective method of measuring the density of sediment that is transported under pressure through a pipeline. This measurement involves the testing of the drop of pressure on two vertical sections of a pipeline, which have the same length and diameter. The technique was tested on the Żelazny Most TSF in southwest Poland, which is a landfill for tailings from the copper ore enrichment process. During the analyzed period of the operation of the device, a medium with a density of 1.379 Mg/m^−3^ to 1.115 Mg/m^−3^ was pumped. The used density measurement technique enables the density to be measured with a slight delay, which results from the time in which the slurry flows from the dredging pump to the outlet of the densimeter, i.e., about 10 s. The accuracy of the measurement with the use of the proposed device was assessed by taking sediment samples from the measuring system and measuring their weight and volume. Comparison of the results from the hydraulic densimeter with the results obtained from the reference technique shows the acceptable accuracy of determining the density of the dredge spoil for the purpose of controlling the dredging process. The relative error of the density measurement with regards to the reference method did not exceed 2.5%, and the mean absolute error value was equal to MAE = 0.018 Mg/m^−3^. In addition to the simplicity and accuracy of the proposed method, its advantage is also the ease of installing measuring apparatus on the decks of small dredging units, as well as the low cost of using the device when compared to other available density measurement techniques.

The disadvantage of the device, however, is its sensitivity to rapid changes in the concentration of the pumped slurry. The temporary flow of liquids of different density through both arms of the device translates into large pressure changes in manometers and significant fluctuations in density values. Similar instabilities occur during the start-up, stoppage and standstill of the dredging pump. Such unstable readings are easily identified and removed during the analysis of results. The authors, for this purpose, used the results of a simultaneous flow measurement with an electromagnetic flow meter. Moreover, the results of the density measurements show slight fluctuations, which could be due to temporary changes in the shear resistance during the movement of the sediment in the pipeline. Pulsations can be removed by filtering the results. The authors successfully applied the wavelet filter of the Block James–Stein estimator. The analysis of the pulsation frequency of the pressure gauge indications using the FFT method did not show the existence of harmonic components in the obtained measurement signal, which would indicate the influence of external factors on the densimeter reading (from vibrations of the pump system or the vessel’s hull).

## Figures and Tables

**Figure 1 sensors-22-07857-f001:**
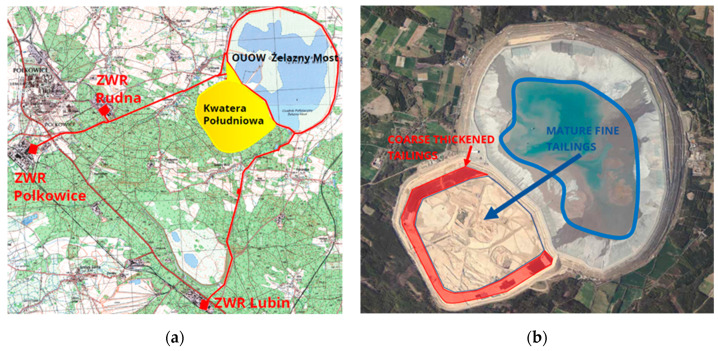
(**a**) Hydrotransport network map; (**b**) satellite image of TSF Żelazny Most (geoportal).

**Figure 2 sensors-22-07857-f002:**
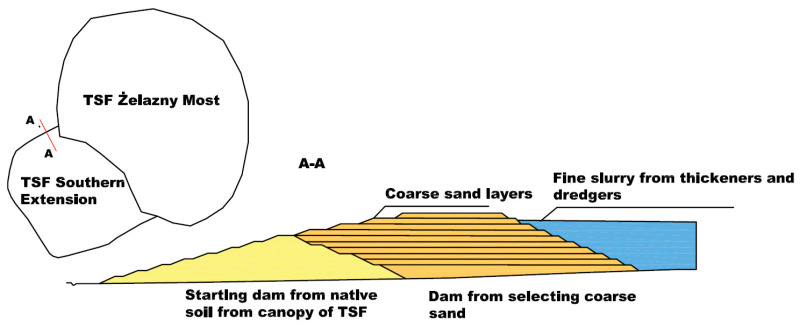
The cross-section of the dam of the South Extension (SE). The distribution of structural material in the body of the dam.

**Figure 3 sensors-22-07857-f003:**
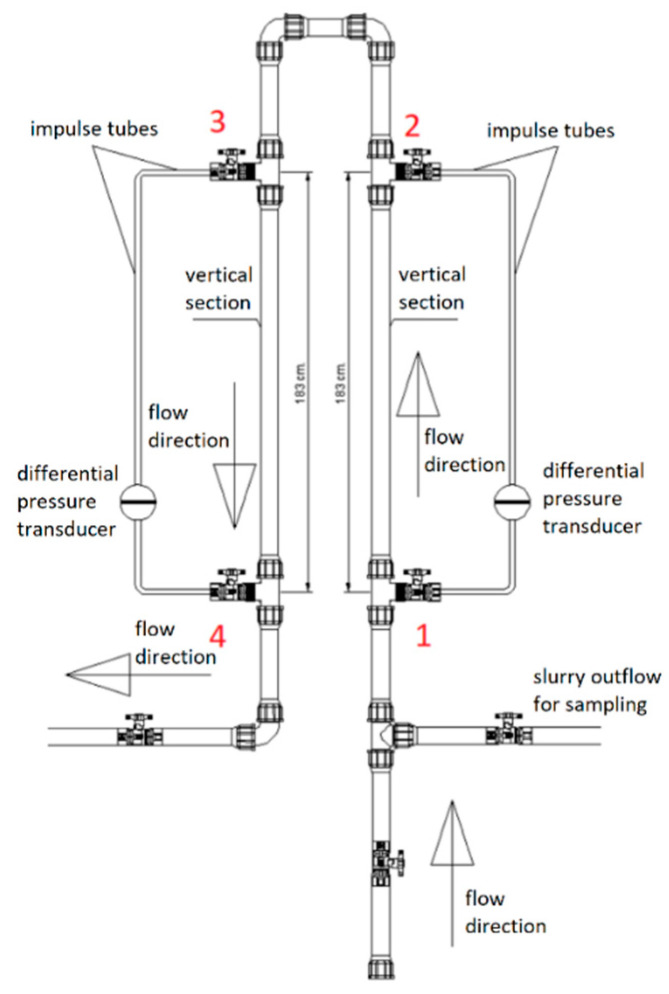
Scheme of the installation of the hydraulic densimeter. 1–4 pressure sampling points.

**Figure 4 sensors-22-07857-f004:**
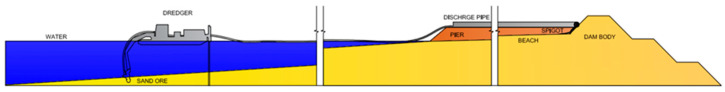
Scheme of the system showing the depositing of the dredge spoil on the beach.

**Figure 5 sensors-22-07857-f005:**
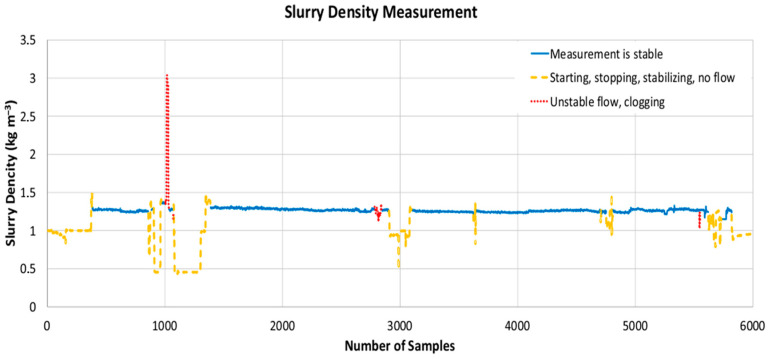
Results of the sediment measurements with the use of the hydraulic densimeter.

**Figure 6 sensors-22-07857-f006:**
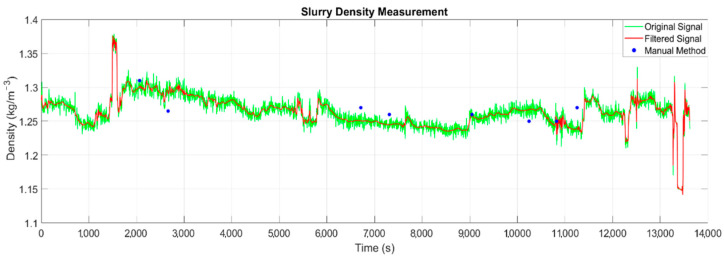
The results of the sediment density measurements over time: original signal (green line); signal after filtering with a wavelet filter (red line); results of manual measurements (blue dots).

**Figure 7 sensors-22-07857-f007:**
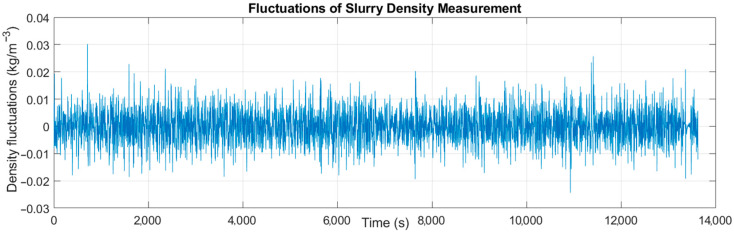
Fluctuations of the slurry density measurement.

**Figure 8 sensors-22-07857-f008:**
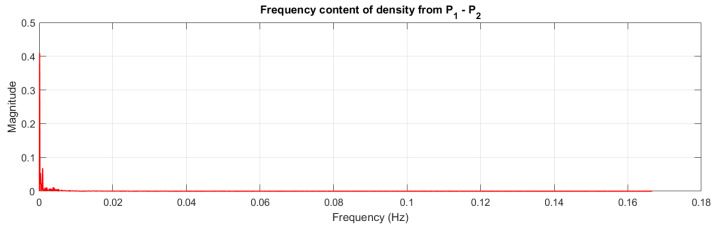
Frequency analysis of the density measurements using FFT.

**Table 1 sensors-22-07857-t001:** Methods for measuring the density of a solid–liquid mixture.

Principle of Measurement	Description	Advantages	Disadvantages
Methods based on mass measurements	The volumetric method consists of selecting a sample for a vessel of a given volume and then weighing the sample.	Low costs of the tests, simplicity of the tests, simple equipment, high measurement accuracy.	No possibility of continuous measurements, difficulties in taking a reliable sample.
Methods based on pressure measurements	The method consists of measuring the pressure difference between two measuring points while taking into account hydraulic losses [12,13,14,15].	Low costs of the tests, simple measuring equipment, the possibility of conducting a continuous measurement.	Measurement points should be placed at different ordinates of the system; requires a homogeneous medium and a steady flow.
Methods based on the measurement of forces (displacements); mass Coriolis flowmeters	The determination of the size of the force (by measuring the displacement) allows the mass flux to be determined [16]. Additionally, by examining the frequency of the wave, it is possible to determine the density of the medium on the same device.	It is one of the most accurate methods that is used to calibrate other methods; the possibility of conducting a continuous measurement of several parameters.	The high cost of the test due to the technically advanced measuring system, which requires a section that is able to be set into vibration. This is difficult for large installations and when on a vessel.
Methods based on acceleration measurements	In a separate section of the pipeline, a specific force is generated and the acceleration caused by it is measured. This allows the density to be determined [17,18]. Mass concentration type method.	The possibility of conducting a continuous measurement; a wide range of diameters of pipes that can be used in measurements.	The high costs of testing a new method; a small network of distributors and maintenance services; the device is placed between the pipeline’s sections.
Methods based on radiation	By having the source parameters, the share of radiation absorbed by the medium is examined using radiation intensity [19,20,21,22].	Widely used; a large network of distributors and maintenance services; the possibility of conducting a continuous measurement; small dimensions of the device when compared to other methods; non-invasive in relation to the pipeline’s network—the possibility of installing on the pipeline.	The high cost of the test; considered to not be environmentally unfriendly due to the use of radioactive elements; requires separate permits and the application of appropriate procedures during its use.
Methods based on vibrations	A section of the pipeline is made to vibrate, the parameters of which are recorded. Based on the frequency of the vibrations, the density is determined [23].	The possibility of conducting a continuous measurement; the relatively low cost of measurements.	A new method; a small network of distributors and maintenance services; devices installed between the sections of the pipeline; requires the dimensions of the pipe that can make it vibrate.
Methods based on ultrasounds	The method is based on the measurement of the ultrasonic wave reflection at the boundary of the media [24,25,26,27,28]	The possibility of conducting a continuous measurement; small dimensions of the device; requires installation between the pipeline’s sections of a ring that does not interfere with the pipeline’s cross-section.	High costs of the tests; requires relatively frequent, but simple calibration.
Methods based on electromagnetic	Part of industrial process tomography, most popular based on electrical properties are electrical capacitance tomography (ECT), electrical resistance tomography (ERT) and electromagnetic tomography (EMT). Tomography is a cross section thru pipe, using sequence of picture can give information about flow include density. Volumetric concentration type method [29,30,31].	Give additional information of flow like flow regime, using correlation of images velocity profile. Non-invasive or low-invasive in to the pipe.	A new method, moderate spatial resolution of the image, need to use high-performance computing.
Methods based on conductivity	Similar to ERT method, using relation between conductivity and volumetric concentration [32].	Quick response sensor, non-invasive	A new method, high dependance of temperature, need for precision calibration on particular mixture.

**Table 2 sensors-22-07857-t002:** Results of the analysis of the unstable operation of the densimeter.

Description	Measurement Numbers
Measurement is stable	381-856, 883-903, 969-1000, 1032-1066, 1387-2780, 2841-2902, 3094-3619, 3642-4701, 4709-7750, 4808-5541, 5549-5617, 5730-5817
Starting, stopping, stabilizing, no flow	1-380, 857-882, 904-968, 1073-1386, 2093-3093, 3620-3641, 4702-4708, 4751-4807, 5618-5729, 5818-6116
Unstable flow, clogging	1002-1031, 1067-1072, 2781-2840, 5542-548

**Table 3 sensors-22-07857-t003:** The results of the density measurements using the manual method and the comparison with the results obtained from the apparatus (hydraulic densimeter).

Time	Manual Measurement	Measurement Using the Developed Apparatus ^1^	Error in %
11:50:00	1.310	1.301	0.70
12:00:00	1.265	1.292	−2.15
13:20:00	1.270	1.250	1.54
13:30:00	1.260	1.245	1.21
14:00:00	1.260	1.259	0.06
14:20:00	1.250	1.267	−1.38
14:30:00	1.250	1.233	1.38
14:40:00	1.270	1.239	2.44

^1^ Values after filtering $\hat{\rho }$.

## Data Availability

We do not report any data publicly available.

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
