# Peer review of "A Simple and Effective Method for Measuring the Density of Non-Newtonian Thickened Tailings Slurry during Hydraulic Transport"

_sensors, 2022, doi:10.3390/s22207857_

Round 1

Reviewer 1 Report

MAJOR COMMENTS

 It must be made clearer, which equipment/methodology is reported in the paper. Abstract states that “The article presents a prototype installation for measuring the efficiency of the solid phase of the "in situ" dredging process in real time. … The installation consists of a flow meter, a densimeter, and a section for measuring the head loss of the flow of the slurry. The applied methodology allows for the current assessment of the dredger's operating parameters ….” However, the article focusses on measurement of slurry density only (“the hydraulic densimeter’), not on the entire installation. For instance, no attention is paid to the role of a flowmeter in the installation (the flowmeter is not required to measure the slurry density using the hydraulic densimeter).

 Survey of measuring methods for slurry density in a pipe must be more complete (e.g. the conductivity method is missing in Table 1) and more thoroughly described. This includes proper references to relevant literature. The existing references are insufficient (more relevant references must be added), inaccurate (some of the existing references are incomplete and cannot be used to find the literature source) and wrongly used in the manuscript (e.g., [1] dies not discuss new safe storage technologies, [4] does not discuss the beach depositing process … and there are many more mismatches in other references).

 Some suggestions for appropriate references:

 U-tube densitometer principle and applications including field application:

Wilson K.C., Addie G.R., Sellgren A. and Clift R. (2006). Slurry Transport using Centrifugal Pumps. 3rd Edition. Springer, USA.

 Radiometric measurement principle and use of nuclear densitometer in a laboratory pipe:

 Radiometric measurement principle and use of nuclear densitometer in a laboratory pipe:

Krupička J, Matoušek, V (2014) Gamma-ray-based measurement of concentration distribution in pipe flow of settling slurry: Vertical profiles and tomographic maps. Journal of Hydrology and Hydromechanics 62: 126–132

Use of nuclear densitometer in dredging applications:

van den Berg, C. (2013) IHC Merwede Handbook for Pumps and Slurry Transportation

Field non-nuclear densitometer Alia:

Peters, J. (2022), “Innovative approach to production monitoring and process control with ADM nonnuclear density meter”, Proceedings of the WEDA Dredging Summit & Expo ‘22, Houston, TX, USA, July 25-28, 2022. (the conference paper rather than the company commercial publication referred to in the manuscript).

 Pressure sensors and conductivity probes:

van Wijk, J.M., de Hoog, E., Talmon, A.M., C. van Rhee (2022). Concentration and

pressure measurements of dense sand and gravel multiphase flows under transient flow conditions in a vertically oriented closed conduit - assessment of system and sensor performance, Flow Measurement and Instrumentation, 84 (2022) 102126

MINOR COMMENTS AND SUGGESTIONS

 Please, describe how/where is the hydraulic densimeter mounted to the dredge pipeline (comment on the differences in diameters of the densimeter U-tube and of the dredge pipeline and possible consequences of the differences).

 Figure 5 – use colors for different lines

 Line 144 – non-Newtonian SLURRY (not liquid)

 Line 223 - the phase slip between the SOLIDS (not the mixture) and the flowing liquid

Reviewer 2 Report

This paper develops a set of pressure based mud concentration measurement system around the mud concentration measurement problem in the process of tailings transportation. The experiment shows that the measurement error of the system is within the acceptable range of the project. At the same time, the system has a simple structure, which well solves this difficult problem. However, there are some problems, so it is recommended to modify it.

 1.Part 165-177 introduces some physical characteristics and main components of tailings, but what is the relationship between these characteristics and components and the selected scheme?

2. Table 1 introduces the mainstream non-invasive measurement methods, but lacks a summary of ECT / ERT / EMT technology of electromagnetic methods.

3. The measurement of mud concentration is closely related to the influence of pipe diameter, pipe wall thickness and flow velocity. It is suggested to explain the reason for selecting this method for measurement?

4. The mud concentration in the pipeline can be divided into volume concentration and mass concentration. Please describe various measurement methods in Table 1.

5. Please describe the relationship between formula (1) and formula (2) in detail.

6. Why does the parameter L in formula (2) refer to the fluid velocity.

7. Whether the time delay is considered and corrected during the side measurement process, how long the fluid flows from 1 to 4, and the specific parameters of the measuring device, such as the distance from point 2 to point 3.

8. It is recommended to add the working conditions in the measurement process, such as the flow rate in the pipeline, the speed of the mud pump, etc.

Round 2

Reviewer 1 Report

Revisions carefully done.

Author Response

The answer is in the attached file.

Reviewer 2 Report

The revision of articles has reached the level of periodical publication. At present, the following problem still need to be supplemented :

Table 3 indicates that the measurement error is within 2.5%. At present, there are only eight groups of data at 11:50-14:30. Some of these data have an interval of 10 minutes, some have an interval of 20 minutes and some have an interval of 30 minutes. Why are the intervals inconsistent? Can you supplement data at other stages.

Author Response

The answer is in the attached file.
